# Resolving kinesin stepping: one head at a time

Willi L Stepp[1] , Zeynep Ökten[1,2]

Kinesins are well known to power diverse long-range transport processes in virtually all eukaryotic cells. The ATP-dependent processive stepping as well as the regulation of kinesin' activity have, thus, been the focus of extensive studies over the past decades. It is widely accepted that kinesin motors can self-regulate their activity by suppressing the catalytic activity of the "heads." The distal random coil at the C terminus, termed "tail domain," is proposed to mediate this autoinhibition; however, a direct regulatory influence of the tail on the processive stepping of kinesin proved difficult to capture. Here, we simultaneously tracked the two distinct head domains in the kinesin-2 motor using dual-color super resolution microscopy (dcFIONA) and reveal for the first time their individual properties during processive stepping. We show that the autoinhibitory wild-type conformation selectively impacts one head in the heterodimer but not the other. Our results provide insights into the regulated kinesin stepping that had escaped experimental scrutiny so far.

## Introduction

Maintenance of a eukaryotic cell is a daunting task of logistics. One key organizer of the eukaryotic cytoplasm is kinesin, a microtubule-associated molecular motor that transports cargo in diverse settings throughout the cell (1, 2, 3, 4, 5, 6). After association with the trail, kinesin takes many steps in a hand-over-hand fashion with its two head domains and covers micrometer distances in vitro (Fig 1A, top panel). To this end, the motor is propelled by the energy provided by two alternating ATP hydrolysis cycles in the so-called "head" domains (7, 8). Communication between the respective cycles ensures that at least one head remains bound to the microtubule to prevent premature dissociation of the motor from its track (9, 10). The timing of these cycles is characterized by the so-called dwell times, for example, the time one head remains bound to the filament between steps (11).

Notably, the ATPase activity of kinesin motors can be suppressed by a self-regulatory mechanism termed autoinhibition (12, 13). This is thought to be achieved by folding of the distal C-terminal tail domain onto the N-terminal head domains (Fig 1A, bottom panel). Either removal of the distal tail or preventing the inhibitory folding relieves

autoinhibition in vitro (Fig 1A, top panel) (12, 14). Indeed, autoinhibition is proposed to interfere with the entry into a "run" as well as with the stepping of the motor (8, 9, 10, 12, 13, 14, 15, 16). Competitive binding to cargo or phosphorylation are thought to disengage the tails from the heads in vivo and in vitro (15, 16, 17 Preprint). Importantly, ectopic activation has been shown to considerably hamper kinesin function in vivo, suggesting that self-inhibition is integral to kinesin-dependent transport processes (18). How the tail-mediated inhibition interferes with the dynamic stepping of a kinesin motor at the molecular level remains an open question (17 Preprint, 19, 20).

Several kinesins that belong to the kinesin-2 family form heterotrimeric complexes comprising two distinct motor subunits and one nonmotor subunit (15, 21, 22, 23). In the example of the heterotrimeric KLP11/20/KAP motor from *Caenorhabditis elegans*, we previously unmasked the distinct contributions of the KLP11 and KLP20 subunits to the motility and autoinhibition of the heterodimeric KLP11/20 motor in vitro (24). Indeed, the presence of two kinetically distinct head domains in a kinesin motor long provoked the question of "limping" during the stepping cycles, that is, a difference in the stepping behavior of the two heads (17 Preprint, 19, 25, 26, 27, 28, 29, 30, 31). Limping in kinesin-1 could be enforced under load; however, limping has so far not been resolved with full-length wild-type motors during unperturbed stepping (32). Resolving limping, in particular, necessitates the separation of the dwell time information for each individual head domain within the dimeric motor.

Here, we have used the heterodimeric nature of the KLP11/20 motor to extract the dwell times from each distinct head simultaneously. To this end, we implemented dual-color fluorescence imaging with one-nanometer accuracy (dcFIONA) that exposed for the first time the respective dwell times of individual head domains during stepping. The capability to extract information simultaneously from both heads ultimately confirmed the previously suggested limping behavior as well as the inhibitory impact of the tail domain on the stepping of the kinesin motor.

## Results and Discussion

### Dual-color step detection with differentially labeled kinesin-2

To follow the two head domains independently, we introduced SNAP- and Halo-tags at the N termini of the KLP11/20 heterodimer

[1]Physik Department E22, Technische Universität München, Garching, Germany   [2]Munich Center for Integrated Protein Science, Munich, Germany

Correspondence: zoekten@ph.tum.de

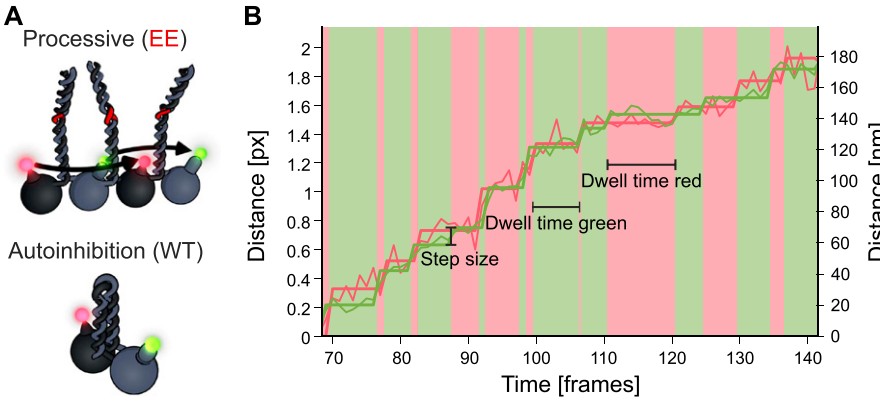

**Figure 1. dcFIONA setup allows concurrent step detection of both heads in kinesin-2.**
**(A)** (top) Depiction of the presumed asymmetric hand-over-hand stepping mode with a heterodimeric kinesin-2 that is labeled with two different fluorophores on its respective head domains. (bottom) Illustration of autoinhibition with the C-terminal tail folded back onto the head domains that in turn suppresses the ATPase activity of the motor. **(B)** Overlaid stepping of the eeKLP11[Halo] and the eeKLP20[SNAP] head domains are shown in green and red, respectively. Data were collected during stepping on microtubules at 0.4 μM ATP. Alternating movement of the motor domains can be seen with corresponding, color-coded dwell times highlighted in the background. A spatial mapping of one channel to the other was not performed.

with wild-type stalk (wtKLP11/20 hereafter) and the construct that contained activating mutations in the stalk (KLP11G451E; S452E/ KLP20G444E, G445E; eeKLP11/20 hereafter), respectively (Fig 1A). The corresponding fluorescent Janelia Fluor dyes of the SNAP- and Halo-tags (JF646 color coded red, JF549 color coded green) labeled the KLP11 and KLP20 subunits with exclusive specificity (33, 34) (Fig S1). Motors labeled in this way showed the expected run lengths (Fig S2).

Using our custom-built setup (21), which we now extended with an additional channel (see the Materials and Methods section), we performed dcFIONA experiments to track both heads at the same time with exact temporal relation and nanometer resolution. At limiting ATP concentrations (0.4 μM), we resolved the stepping of each head individually (Fig 1B). As expected, the step size of the dual-labeled eeKLP11[Halo]/20[SNAP] was consistent with our previous findings with the eeKLP11[Halo]/20 motor that was labeled on one head domain only (13.2 and 13.9 nm versus 13.4 nm (21)) (Fig S3).

## The two heads of the KLP11/20 motor display distinct stepping behaviors

The dwell times for kinesin constructs that were labeled on one head only were shown to be distributed according to a convolution of two exponentials (7, 21). Only the steps of the labeled head can be observed in these experiments, whereas the step of the other head is "hidden." As the hidden step has to occur before the next observed step is possible, two rate-limiting events are necessary for each observed step. This leads to a convolution of two exponentials for the dwell time distributions in these experiments (7, 21). In our measurements, we can now extract the dwell times in the "step primed" position, that is, only the time a head spends in the trailing position before it takes the step.

At limiting ATP concentrations, we measured the individual dwell times of the two heads in the eeKLP11[Halo]/20[Snap] motor (Fig 1B). For these dual-color measurements, the convolution of two exponentials is expected to split into one single exponential distribution for each head (35). Intriguingly, however, we observed two different distributions (Fig 2, left versus right panels). Whereas the dwell times obtained from the KLP20 head domain displayed a single exponential distribution as expected, the dwell times extracted from the KLP11 head domain clearly deviated from a single

exponential but were instead consistent with a convolution of two exponentials (Fig 2, right panels).

To exclude any influence of the respective fluorophores or their recording by our setup, we switched the dyes (eeKLP11[Halo]/20[SNAP] versus eeKLP11[Halo]/20[SNAP]) on the respective head domains (Fig 2A versus B). In addition, we also switched the position of the Halo- and SNAP-tags themselves (eeKLP11[Halo]/20[SNAP] versus wtKLP11[SNAP]/ 20[Halo]) to exclude any influence of the specific tags on the behavior of the motor per se (Fig 2B versus C), as well as the relative positions of the KLP11 and KLP20 head domains (Fig S4A and B). In all cases, we confirmed the dwell time distribution as a convolution of two exponentials for the KLP11 head domain (Fig 2, right panels), whereas the KLP20 head domain consistently displayed a single exponential distribution (Fig 2, left panels).

To further test the consistency of this observation, we fitted both data sets using the same convolution of two exponentials with independent parameters (see Supplementary Information). A big ratio of the parameters $k_2/k_1$ for this model leads to a distribution that is close to a single exponential, whereas a fit with a ratio of ~1 features the distinct fall-off at short dwell times. For the KLP11 head domain, the ratio of the two involved parameters was close to 1 (Fig 2, right panels), indicating a similar influence of both rate-limiting events on the stepping behavior. For the KLP20 head domain, in contrast, the ratio was about 100-fold higher, ultimately resulting in a near single exponential fit (Fig 2, left panels). We also performed more advanced statistical tests on these fits showing the difference of the two distributions at the 99% confidence level (Fig S5). This is a strong indicator that no second rate-limiting step influences the stepping of KLP20.

For the KLP11 head, however, these findings suggest that the steps taken include another rate-limiting event in addition to the waiting time for ATP binding (7). The observed differences in the dwell time distributions as displayed by the KLP11 and KLP20 head domains ultimately confirm the presumed limping for hetero-dimeric motors (17 Preprint, 19, 25, 26, 27, 28, 29, 30, 31). This behavior of a single head domain could so far not be resolved by tracking the net movement of the motor because of the similar mean dwell times of the two heads (25).

What is the origin of the other rate constant that is displayed specifically by the KLP11 head domain? Notably, our previous work

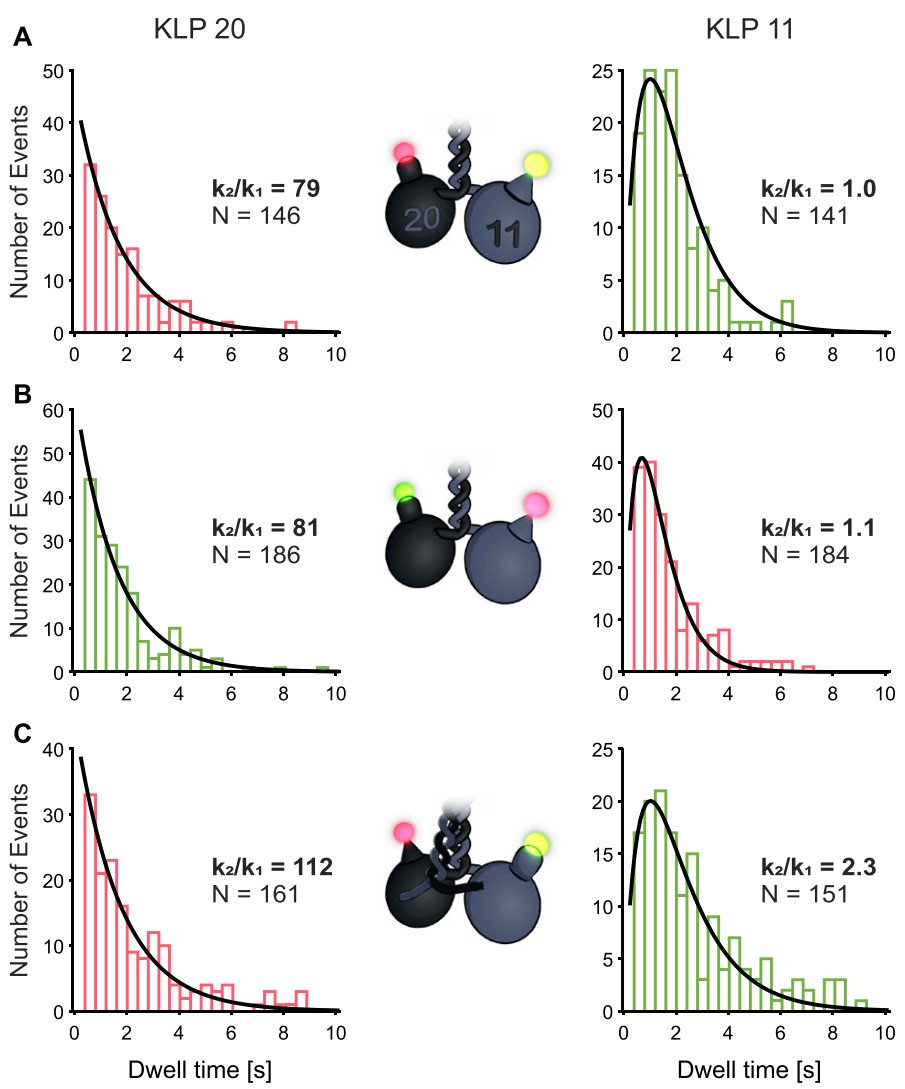

**Figure 2. Dwell time distributions of the KLP11 and KLP20 head domains are different.**
**(A, B, C)** Dwell time distributions of KLP20 and KLP11 are depicted with bars in the respective colors (green for JF549 and red for JF646) the data were collected in. The width of the bins represents the cycle time (405 ms); therefore, the fits are expected to be independent of the binning. Fits performed with a convolution of two exponentials with same settings and starting point (see Supplementary Information for details). KLP20 dwell times show a distribution close to a single-exponential distribution (left panels). KLP11 dwell times are fitted well by the convolution of two exponentials with similar values for both parameters $k_1$ and $k_2$ (right panels). Fitting of the KLP20 data shown in the left panels with the same model yields a ratio of the two parameters that is about two orders of magnitude higher, resulting in a quasi-single exponential fit. All fits resulted in $r^2$ values >90%. N is the number of steps analyzed, n is the number of motors included in the analysis. **(A)** eeKLP11[Halo]/20[SNAP] (20: $k_1 = 0.6$ s$^{-1}$, $k_2 = 47.2$ s$^{-1}$; 11: $k_1 = k_2 = 1.0$ s$^{-1}$; n = 11). **(B)** eeKLP11[Halo]/20[SNAP] (20: $k_1 = 0.6$ s$^{-1}$, $k_2 = 48.5$ s$^{-1}$; 11: $k_1 = 1.4$ s$^{-1}$, $k_2 = 1.5$ s$^{-1}$; n = 11). **(C)** wtKLP11[SNAP]/20[Halo] (20: $k_1 = 0.6$ s$^{-1}$, $k_2 = 67.2$ s$^{-1}$; 11: $k_1 = 0.6$ s$^{-1}$, $k_2 = 1.4$ s$^{-1}$; n = 23). See Fig 3 for more detail on the influence of the wt-versus ee-version of the motor. Fitting function: $A(e^{-k_1 t} - e^{-k_2 t})$.

with the wtKLP11/20 suggested an asymmetric autoinhibition mechanism (24). It required both the presence of the tail and the correct positioning of the KLP11 head within the wtKLP11/20 heterodimer. Strikingly, however, solely swapping the positions of the KLP11 and KLP20 head domains sufficed to activate the autoinhibited wtKLP11/20 motor in single molecule and bulk ATPase assays (24, 36). These findings provoke the question whether the presence of the C termini in the eeKLP11/20 per se influences the dwell time distribution of the KLP11 head domain. If true, the autoinhibitory folding in the wtKLP11/20 stalk would be expected to enhance this influence specifically in the KLP11 data (Fig 1A, bottom panel).

### Difference in stepping gives insight into the autoinhibition of kinesin-2

To test this hypothesis, we extracted long dwell times (>2 s) from the KLP11 distributions from Fig 2 (right panels) and refitted them with a single exponential model (Fig 3A). This was necessary because the

convolution of two exponentials with similar parameters is not very stable on the fit of the actual parameters but relies more on the ratio of the two. The exact parameters are, therefore, not very accurate, and we decided to truncate the feature that is characteristic for the convolution to obtain stable fits for the single exponential dwell times.

For the wtKLP11/20 motor, the resulting dwell time parameter increased 1.6-fold when compared with the eeKLP11/20 that contains the ATPase-activating mutations in the stalk (Fig 3, left panels). This 1.6-fold difference is in fact consistent with the decreased speed of the wild-type motor at saturating ATP concentrations (Fig 3B) (20).

We previously demonstrated that simply switching the relative positions of the KLP11/20 heterodimer is also sufficient to relieve the autoinhibition without the necessity to mutate the stalk domain. The latter implicates an asymmetry autoinhibition mechanism in the heterodimeric KLP11/20 (Fig 1) (24, 36). If the increased dwell time of the KLP11 head domain indeed results from an asymmetric inhibition by the stalk/tail, switching the position of the KLP11 head with the KLP20 alone would be expected to shorten the

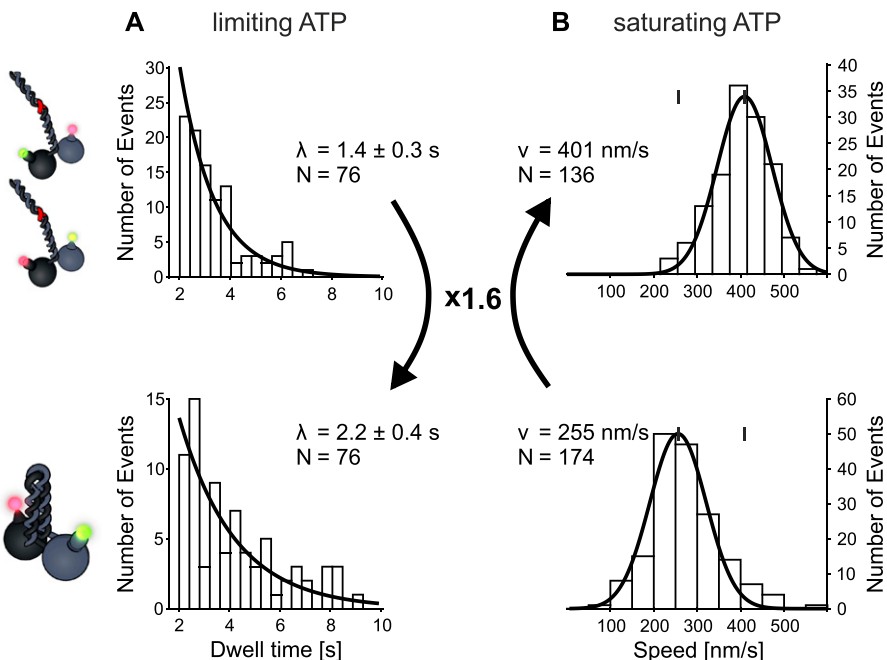

**Figure 3. Presence of the wild type C termini selectively prolongs the dwell times in the stepping of the KLP11 head domain.**

**(A)** Truncated dwell times over 2 s of KLP11 fitted with a single-exponential model (data from Fig 2 right panels, A + B eeKLP, C wtKLP). The ratio of the dwell times from the wtKLP11/20 (bottom, 2.2 s) to the eeKLP11/20 (top, 1.4 s) is 1.6. This factor of 1.6 is also consistent with the ratio of mean dwell times as seen in Fig 2 (right panel, [A] + [B] eeKLP, [C] wtKLP). Fits were performed using a maximum likelihood approach with a truncated distribution and resulted in pseudo $r^2$ values >90%. **(B)** Comparison of velocities from eeKLP11/20 (top, from (21), $\mu$ = 90 nm/s) and the wtKLP11/20 (bottom, $\mu$ = 64 nm/s, $r^2$ = 96%) at saturating ATP concentrations. The ratio of speeds (eeKLP11/20: 401, wtKLP11/20 255 nm/s) is the reversed value of the ratio of dwell times (see Fig S2 for run length data of the respective motors).

dwell times of the KLP11. Strikingly, despite the presence of the wild-type stalk/tail, swapping the positions of the KLP11 and KLP20 heads indeed sufficed to reduce the dwell time of the KLP11 head domain (Fig S4C) (24, 36). Specifically, previous ATPase and filament gliding experiments showed an activity of the wtKLP11-20/20-11 construct to be roughly between the wtKLP11/20 and the eeKLP11/20. We compared the dwell times and velocity of this motor to the wtKLP11/20 as done previously for the eeKLP11/20. Consistent with the finding that it shows intermediate activity in ATPase assays (24), the activity ratios compared with the wtKLP for both speed and KLP11 dwell time are 1.24, higher than the wtKLP but below the values for the eeKLP11/20 (Figs 2 and S4C and D).

We think that the same modulation in the head–tail interaction is responsible for the observed differences at both limiting and saturating ATP concentrations. This manifests in shorter dwell times and higher speeds for the mutated motors (eeKLP and switched heads) compared with the wtKLP. An influence of the head–tail interaction on the ADP release time in the ATP-hydrolysis cycle could explain both effects.

The rate-limiting event present in both dwell time distributions is attributed to the ATP waiting time at low ATP concentrations (7). Based on previous data (37, 38, 39), we speculate that the other rate-limiting event in the dwell time distribution of the KLP11 head domain is the tail-suppressed ADP release. This effect is strong in the wild-type motor in which the flexible hinge in the stalk enables autoinhibitory folding and consequently enhances the "head–tail" interaction (Fig 1A, bottom panel). When the stalk is mutated to prevent autoinhibitory folding (Fig 1A, top panel), the head–tail interaction is hampered, thus selectively shortening the dwell times in the KLP11 stepping (Fig 2, right panels A + B versus C). Intriguingly, this shortening is also observed *in the presence* of the inhibitory wild-type stalk/tail domain when the positions of the head domains were swapped (Fig S4). The latter exposes the asymmetry in the autoinhibition mechanism and shows that for an

efficient stalk/tail-mediated inhibition, the KLP11 head domain must be in its wild-type configuration.

Taken together, our capability to distinguish between the two head domains during processive stepping provides compelling support for an asymmetric autoinhibition mechanism in the KLP11/20 heterodimer (24). Indeed, our results unmask for the first time an influence of the configuration of the stalk on the dynamic stepping of a physiological kinesin motor.

Previous tracking of one head domain in the homodimeric kinesin-1 at nanometer resolution using FIONA already represented a major breakthrough, given the small 8-nm net displacement of the motor (7). Being able to trace two head domains simultaneously with the dcFIONA introduced here now allows the dissection of the *specific* contributions of the head domains to the processive stepping and the regulation thereof at the single molecule level. The next major experimental challenge towards a comprehensive understanding of the kinesin stepping mechanism will be the correlation of the stepping behavior to specific events in the respective ATPase cycles of the motor domains.

## Materials and Methods

### Constructs and design

All constructs were based on the heteromeric kinesin-2 KLP11/20 active in the intraflagellar transport in *C. elegans*. eeKLP mutations were performed as described previously (26). Halo- and SNAP-tags were fused to the N terminus of the respective sequences where applicable. The constructs used are as follows:

- eeKLP11[Halo]
- wtKLP11[SNAP]

· eeKLP20$^{SNAP}$
· wtKLP20$^{Halo}$
· wtKLP11-20$^{SNAP}$
· wtKLP20-11$^{Halo}$

### Protein expression, purification, and fluorescent labeling

All proteins were expressed and purified as described previously (21). For fluorescent labeling, Janelia Fluor dyes JF549 and JF646 in Halo- and SNAP-conjugated variants were used (33). The dyes were mixed in 1:1 ratio before incubation, and the incubation time with the dyes was prolonged to 90 min.

### Microscope setup

Single-molecule experiments were performed on a custom-built setup described previously (21). A 555-nm laser (Oxxius) was added to the setup as well as a color split/recombine setup using a high- and a low-pass dichroic to offset the channels on the camera chip.

### Single-molecule experiments

Speeds and run length were measured at an ATP concentration of 2 mM. Movies were recorded with an exposure time of 200 ms, and 500 frames were recorded before changing the position in the sample.

For step detection experiments, the ATP concentration was reduced to 0.4 μM; the creatine phosphate/creatine phosphokinase system guaranteed stable ATP concentrations over the duration of data collection. Movies were recorded with an exposure time of 400 ms for dual color experiments, resulting in a cycle time of 405 ms.

### Data analysis

All data analysis was performed using ImageJ and custom routines implemented in MATLAB (Mathworks Inc.). Traces for speed and run length measurements were extracted by identifying and following peaks depending on their brightness. A position with subpixel accuracy for these traces was assigned using a radial center approach (40). Runs over several frames were connected by following peaks according to their distance to a peak in the previous frame. Overall distances were calculated with respect to the first detected position in a run. Speeds where then calculated by performing a linear regression on the distance over time data and extracting sequences that fitted with an $r^2$ value higher than 95%. Run lengths were determined from the maximum distance from the starting point for each run.

For step detection experiments, a least-squares fit procedure was used to fit a two-dimensional Gaussian profile to the peak data with a starting point deduced from the initial detection of the brightest pixel. This fit provided a higher accuracy subpixel position for each frame, compared with the radial center approach. Peaks were followed frame by frame again and the distance to the position in the first frame was recorded to a time−distance trace. Because of the lower speeds, the distance over time traces show distinct relocation events for each step of the respective head with plateaus in between when the head was bound to the filament. An implementation of the Potts algorithm was used to detect the underlying stepping from the distance over time trace (41). The used Potts parameter was 0.3 times the SD of the run that a smoothed version of the same run with a window of 20 frames (*smooth*(*run,20*) function in MATLAB) was subtracted from. Single-position spikes in the detected stepping trace were filtered out. The individual sizes of steps were calculated from the mean distances from the original position before and after each step.

Sequences with alternating stepping from one head and the other were extracted from the whole runs. Dwell times for one head were then calculated from the time of a step in the other color to the next step in the heads color. Results were plotted in histograms with a bin width which was the cycle time of the experiment. This ensures no influence of the binning on the resulting fits. The MATLAB routines and raw data used in this study are available as supplementary files (Supplemental Data 1 and 2).

### Fit to the convolution of two exponentials

In our first impression, the two dwell time distributions for KLP11 and KLP20 were distributed according to a convolution of two- and a single-exponential model respectively. To support this assumption, we fitted both distributions with a double exponential model and show that one of the parameters vanishes for the KLP20 dwell times. The model used was

$$p = A\bigl(e^{-k_1 t} - e^{-k_2 t}\bigr).$$

We used the same settings and starting point for the fit:

$$A = 150; k_1 = 0.3; k_2 = 10.$$

To focus on the shorter dwell times, where the effect of the double-exponential distribution can be seen the best, we introduced a weight function that describes the weight the corresponding data point contributes to the fit:

$$w = 1 - \frac{1}{1.2(1 + e^{-2t+5})}.$$

The errors from the fit are large for some parameters:

(1) $k_2$ values for the "quasi" single-exponential distributions (KLP20) as the fit is weakly dependent on the actual value of $k_2$
(2) $k_1$ and $k_2$ in the fits that converge to a convolution of two exponentials (KLP11) because the fit does not depend on the actual values but on the ratio $k_2/k_1$

For Fig S5, the fits were additionally performed using the *fitlm* function in MATLAB that also reports the statistical analysis of the fit.

## Supplementary Information

## Acknowledgements

We thank Jonathan B Grimm and Luke D Lavis (Janelia Research Campus, Howard Hughes Medical Institute) for their generous gift of the Janelia Fluor compounds. We thank William O Hancock and Matthias Rief for the helpful comments on the manuscript. The research leading to these results has received funding from the European Research Council (Grant Agreement no. 335623) to WL Stepp and Z Ökten and Deutsche Forschungsgemeinschaft SFB863 to Z Ökten.

### Author Contributions

WL Stepp: conceptualization, data curation, software, validation, investigation, visualization, methodology, project administration, and writing—original draft, review, and editing.
Z Ökten: conceptualization, resources, supervision, funding acquisition, project administration, and writing—original draft, review, and editing.

### Conflict of Interest Statement

The authors declare that they have no conflict of interest.

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
