## [Reviewer comments · Life Science Alliance]

Life Science Alliance

Resolving kinesin stepping: one head at a time

Willi Stepp and Zeynep Ökten

DOI: <https://doi.org/10.26508/lsa.201900456>

Corresponding author(s): Zeynep Ökten, Technische Universität Muenchen

Review Timeline:

Submission Date:	2019-06-13
Editorial Decision:	2019-07-09
Revision Received:	2019-08-22
Editorial Decision:	2019-09-20
Revision Received:	2019-09-27
Accepted:	2019-09-27

Scientific Editor: Andrea Leibfried

Transaction Report:

July 9, 2019

Re: Life Science Alliance manuscript #LSA-2019-00456-T

Dr. Zeynep Ökten
Technische Universität Muenchen
Department of Biophysics
James-Franck-Str.1
Garching D-85748
Germany

Dear Dr. Ökten,

Thank you for submitting your manuscript entitled "Resolving kinesin stepping: one head at a time" to Life Science Alliance. The manuscript was assessed by expert reviewers, whose comments are appended to this letter.

As you will see, the reviewers appreciate the first dual-color tracking analysis for a kinesin motor, and they provide constructive input on how to further strengthen your work. We would thus like to invite you to submit a revised version of your manuscript to us, addressing the individual concerns raised by the reviewers. These seem all straightforward to address, but please do get in touch in case you would like to discuss an individual revision point further.

Thank you for this interesting contribution to Life Science Alliance. We are looking forward to receiving your revised manuscript.

Sincerely,

Andrea Leibfried, PhD

Executive Editor
Life Science Alliance
Meyerhofstr. 1
69117 Heidelberg, Germany
t +49 6221 8891 502
e a.leibfried@life-science-alliance.org
www.life-science-alliance.org

B. MANUSCRIPT ORGANIZATION AND FORMATTING:

Reviewer #1 (Comments to the Authors (Required)):

The manuscript reports the first high resolution two color tracking of a kinesin (previous multicolor experiments have been restricted to dynein and myosins).

The authors use a heterodimeric kinesin-2 to allow each motor domain to be labelled in a different color. The simultaneous tracking allows them to measure the dwell time each motor spends in the

trailing position. They find, surprisingly, that while the dwell times of the KLP20 motor fit a single exponential (only waiting for ATP binding), the dwell times of the KLP11 motor fit a double exponential. They compare an inhibited wt motor with a mutated motor that has less autoinhibition and suggest that the second rate constant is related to the autoinhibition of the tail.

At the core of this manuscript are new findings that are in principal suitable for LSA. However, I feel the manuscript was not currently written for a general audience. I also had some concerns about the conclusions relating to the role of tail-autoinhibition causing the limping of the two heads.

Comments:

1) Page 8 - Line 3: The authors need to mention that not only the position of the tags, but also the motor construct was changed in Fig 2C

2) Figure 1B: distances should be given in nm, not px (pixels?)

3) Figure 2: The cartoon in the middle of the figure should make it clear which construct is being used (eeKLP11/20 or wtKLP11/20).

4) Figure 2: Should the fitted parameters (λ_1 and λ_2) have units?

5) Page 8 - Line 10: Until this point of the manuscript the authors have talked only of a single exponential. In this paragraph they fit a double exponential to the same data. They should explain why they did this and what the results mean.

6) Page 9 - Line 8: Please could the authors specify what "this effect" refers to

7) Page 9 - Line 11: Please could the authors explain why they extracted long dwell times? What is the significance of the observation that the 1.6 fold difference between EE and WT is the same as the motor at saturating ATP?

8) Page 9 - Line 18: The authors suddenly start using the term rate constant. I am left confused between the parameters fitted in Figure 2 (dwell times values or rate constants?). The authors should calculate and comment upon the rate constants from their dwell time data. Are they the same for KLP20 and KLP11 steps? Do the WT and EE motors have the same ATP-waiting time?

9) Page 9 and 10: The authors argue that the tail inhibition is contributing to the difference in behaviour of KLP20 and KLP11 heads. If this is the case then why does the EE mutant still show a double exponential for KLP11? Is it because there is still some autoinhibition remaining? If so the authors should include data on a minimal motor construct that lacks the tail domain. If the tail is not the only cause of the KLP11 double exponential, then the authors should discuss what else could contribute.

Reviewer #2 (Comments to the Authors (Required)):

In the manuscript by Stepp and Okten, the authors directly visualized dynamics of individual head of heterodimeric KLP11/20 motor during processive stepping in using dual color FIONA, which is in good agreement with previous studies. The authors confirmed asymmetric stepping during a processive movement in support of previous studies. The quality of the data is high enough. This manuscript can be published with revisions on the following issues.

1. The ATP waiting time of KLP11 in Fig 2. Have the authors confirmed it experimentally by using different concentration of ATP?
2. In page 9, the authors' discussion reads that the preventing auto-inhibition shortens the dwell time. It is confusing, because auto-inhibition of kinesin usually leads to inhibit binding to microtubules.
3. Quantitative measures for fitting should be reported for example in Fig 3.

Reviewer #3 (Comments to the Authors (Required)):

This manuscript uses dual color FIONA to track the stepping of two distinct motors domains of a kinesin-2 motor. The goal is to reveal whether the two heads step at distinct rates (so-called limping) and whether their stepping is rate limited differently from each other. Finally, they generated a construct that prevents tail-head interactions and determined how hyperactivation of kinesin-2 affects the stepping of its motor domains. The authors convincingly showed that the two heads have distinct stepping kinetics, which is affected separately by the tail. The manuscript is definitely worth publication in LSA. This is the first dual color tracking study of kinesin, so it also presents a technical advance in the field.

I am listing my comments below to help authors prepare a revised manuscript:

1. What is ATP concentration? Mention this in the main text.
2. Page 7, line 12: double exponential decay is usually used for the addition of two double exponential decay functions with separate amplitudes and rates. The authors refer to a convolution of two exponentials, in which two exponential decay functions with the same amplitude are subtracted from each other. At the limit where the rates become equal, this equation becomes the Gamma function. I suggest the authors use "convolution of two exponentials" as this is the right term to evaluate the kinetics of two sequential rate-limiting steps that lead to an observable step.
3. Page 7, line 14: I am surprised to see that the authors have not made the most out of their dcFIONA data. Do the heads strictly walk hand-over-hand? Does the stepping originate from the trailing position, and end up in a leading position? How did they register the two fluorescent channels?
4. Page 7 line 14-16 is vague. Explain more clearly that single color imaging does not detect when the unlabeled head takes a step and its kinetics follows convolution of two rate-limiting steps. If both heads are being labeled, this is no longer the case, and one can detect the stepping time of both heads. Therefore, two color traces can be broken down to detect stepping time of each head, and dwell time between the steps would follow a single exponential decay. This was mentioned in the next paragraph, but it would be more appropriate to mention it all together in a single paragraph.
5. Page 2 line 24: Convolution of two exponentials is not fully convincing, especially in B. The authors need to run F-test to justify that addition of an extra parameter to the fit is justified by the fit itself. The best way to address this would be to lower ATP further or increase the frame rate, in order to detect the initial rise of the convolution function.
6. The left side of Figure 3 is very confusing. Selection of dwell times longer than 2 s is arbitrary and cannot be justified. The right way to do this would be to fit the convolution function and compare those rates from the fit.
7. Page 9, line 18: It is confusing to refer to distinct rates as the first rate and the second rate. What do the authors mean? Are these faster and slower rates from the fit?

8. Page 10, line 5: This conclusion is an overstatement. The authors say that they speculate on a model in a previous page. But here they claim that they provide compelling support. They need to soften this conclusion as the model still remains speculative.

9. Figure 1 legend: The formula used refers to lambda, which is often used as the lifetime of exponential decay, whereas the authors refer to rates (k) in the main text. In addition, lambda is multiplied with x , instead, they should have used t , the symbol of time. So, λx should be replaced with $k t$ to avoid confusion. The authors should also report the number of measurements (number of motors and steps analyzed) and report the errors from the fit for each dataset.

Reviewer #1

The manuscript reports the first high resolution two color tracking of a kinesin (previous multicolor experiments have been restricted to dynein and myosins).

The authors use a heterodimeric kinesin-2 to allow each motor domain to be labelled in a different color. The simultaneous tracking allows them to measure the dwell time each motor spends in the trailing position. They find, surprisingly, that while the dwell times of the KLP20 motor fit a single exponential (only waiting for ATP binding), the dwell times of the KLP11 motor fit a double exponential. They compare an inhibited wt motor with a mutated motor that has less autoinhibition and suggest that the second rate constant is related to the autoinhibition of the tail.

At the core of this manuscript are new findings that are in principal suitable for LSA. However, I feel the manuscript was not currently written for a general audience. I also had some concerns about the conclusions relating to the role of tail-autoinhibition causing the limping of the two heads.

Comments:

1) Page 8 - Line 3: The authors need to mention that not only the position of the tags, but also the motor construct was changed in Fig 2C

Response: In Figure 2C we exclude any influence of the two different protein tags that we used to differentially label the respective motor domains, and we therefore we would like the reader to focus on this particular aspect first before we continue with the stalk/tail aspect of the analysis. We have now cross-referenced the Figure 3 in Figure 2C that addressed the aspect of stalk modification in this particular construct (p. 7 l.12).

2) Figure 1B: distances should be given in nm, not px (pixels?)

Response: We now included a second y-axis on the right with the corresponding distances in nm (p. 5 l.7).

3) Figure 2: The cartoon in the middle of the figure should make it clear which construct is being used (eeKLP11/20 or wtKLP11/20).

Response: As the Figure 2 primarily deals with the stepping characteristics of the two different motor domains along with the necessary controls (i.e. fluorophore (B) and protein-tag swaps (C)), we opted to focus the attention exclusively on the heads first and establish the validity of our constructs. If deemed necessary, we are happy to exchange the current depiction with the following:

4) Figure 2: Should the fitted parameters (λ_1 and λ_2) have units?

Response: This is correct and we have updated Figure 2 accordingly (p. 7 l.9-11).

5) Page 8 - Line 10: Until this point of the manuscript the authors have talked only of a single exponential. In this paragraph they fit a double exponential to the same data. They should explain why they did this and what the results mean.

Response: We now state more clearly the occurrence of the single exponentials and the convolution of two exponentials (see also reviewer #3). The reason for fitting both distributions with the same model has also been explained in more detail. What the results mean for both heads is now phrased more clearly here and in the next paragraph (p.7 l.15-p.8 l.6, p.8 l.17-p.9 l.3).

6) Page 9 - Line 8: Please could the authors specify what "this effect" refers to

Response: We now rephrased to clarify the referral of 'this effect' to the previous sentence (p.9 l.20).

7) Page 9 - Line 11: Please could the authors explain why they extracted long dwell times? What is the significance of the observation that the 1.6 fold difference between EE and WT is the same as the motor at saturating ATP?

Response: We now explain in more detail that the long dwell times are extracted due to the nature of the fit (p. 10 l.1-5) (see also reviewer #3). As we cannot pinpoint the exact mechanism of the inhibitory impact of the tail on the dwell times, we can only speculate on the relationship between these two phenomena (p.11 l.3-8). However, please refer to our new data in Suppl. Figure 4 for further consistency between the dwell times and the velocity of the motor at saturating ATP (p.10 l.24-p.11 l.2 & p.25).

8) Page 9 - Line 18: The authors suddenly start using the term rate constant. I am left confused between the parameters fitted in Figure 2 (dwell times values or rate constants?). The authors should calculate and comment upon the rate constants from their dwell time data. Are they the same for KLP20 and KLP11 steps? Do the WT and EE motors have the same ATP-waiting time?

Response: The difference between the rate constants and the dwell time parameters is now stated more clearly in the figures and the text (p.6, p.7 l.5&9-11) (see also reviewer #3 9.). We report more data on the fits now in Supplementary Figure 5 (see also reviewer #3 6.). The rate constants are the same for KLP20. For KLP11 we see differences that we contribute mostly to the fact that fitting of convolutions of exponentials depends mostly on the ratio of the two coefficients instead of on their actual values (p.10 l.1-5). However, we solely speculate on the exact step influenced in the ATPase cycle based on previous findings.

9) Page 9 and 10: The authors argue that the tail inhibition is contributing to the difference in behaviour of KLP20 and KLP11 heads. If this is the case then why does the EE mutant still show a double exponential for KLP11? Is it because there is still some autoinhibition remaining? If so the authors should include data on a minimal motor construct that lacks the tail domain. If the tail is not the only cause of the KLP11 double exponential, then the authors should discuss what else could contribute.

Response: Unfortunately, the truncated constructs of KLP11/20 that we already tried to express for the revision of our paper in EMBO reports in 2017 have never worked (see published peer review information on that paper). To provide evidence for the asymmetric inhibitory impact of the tail on the motor domains, we now switched the relative positions of the random coil tails in the KLP11 and KLP20 subunits instead of their removal. This swapping has substantially disrupted the robust heterodimerization between the respective motor subunits. Together, these results implicate that the C-terminus of the heterodimeric motor is highly sensitive to mutations.

Instead of swapping the distal random coil tails, we therefore constructed a heterodimeric motor with swapped head domains which is expected to also relieve the proposed asymmetric inhibition (Brunnbauer et al., PNAS 2010). Indeed, swapping the KLP11 with the KLP20 head domain in the presence of *inhibitory wild type* stalk/tail sufficed to shorten dwell times of the KLP11 head as we observed with the tail mutant eeKLP11/20 as we now show in the new Suppl. Figure 4. Our findings now directly demonstrate that the presence of the wild type stalk/tail alone is not sufficient to mediate efficient auto-inhibition, but the head domains have to be in their wild type configuration. The latter is entirely consistent with our previous results from bulk ATPase assays (Brunnbauer et al., PNAS 2010) (p.8 l.12-13, p.10 l.11-p.11 l.2, p.24).

Reviewer #2

In the manuscript by Stepp and Okten, the authors directly visualized dynamics of individual head of heterodimeric KLP11/20 motor during processive stepping in using dual color FIONA, which is in good agreement with previous studies. The authors confirmed asymmetric stepping during a processive movement in support of previous studies. The quality of the data is high enough. This manuscript can be published with revisions on the following issues.

1. The ATP waiting time of KLP11 in Fig 2. Have the authors confirmed it experimentally by using different concentration of ATP?

Response: Working with these motors at low ATP concentrations is challenging. The low end of the window of possible concentrations is given by the fact that motors do not show any activity if the ATP concentration is significantly lower than the used 0.4 μ M and instead remain statically associated with the microtubules. The upper limit is given by our chosen cycle-time that is optimized for the most reliable step detection possible. We worked to optimize the concentration for a reliable detection of steps for KLP11/20 and unfortunately don't see a possibility to work at ATP concentrations that would give a detectable change in the dwell time distributions and allow reliable step detection at the same time.

2. In page 9, the authors' discussion reads that the preventing auto-inhibition shortens the dwell time. It is confusing, because auto-inhibition of kinesin usually leads to inhibit binding to microtubules.

Response: Even though the KLP11/20 motor is 'switched off' as judged from ATPase assays (Brunnbauer et al., PNAS 2010), it is still capable of moving processively on surface attached microtubules. This indeed starkly contrasts the behavior of its in vivo partner Osm-3 kinesin-2 which is not capable of processive movement in its auto-inhibited conformation. However, the wild type motor always moves slower when compared to the activated eeKLP11/20 motor. In fact, Friedman and Vale (Nature Cell Biology 1999) too reported different velocities for different configurations of kinesin-1 that influenced the auto-inhibition of the motor. This means that both, the binding to the microtubules and the velocity when moving on a microtubule are influenced by what we commonly call auto-inhibition.

3. Quantitative measures for fitting should be reported for example in Fig 3.

Response: We now report quantitative measures for the fits in Figure 3 as well as in the figure legend. We also mention why the standard errors are not good measures to evaluate the fits in Figure 2 (see also reviewer #3). Comprehensive statistics to the fits in Figure 2 can now be found in Suppl. Figure 5 (p.12 l.12&14, p.9 l.1-2, p.26).

Reviewer #3

This manuscript uses dual color FIONA to track the stepping of two distinct motor domains of a kinesin-2 motor. The goal is to reveal whether the two heads step at distinct rates (so-called limping) and whether their stepping is rate limited differently from each other. Finally, they generated a construct that prevents tail-head interactions and determined how hyperactivation of kinesin-2 affects the stepping of its motor domains. The authors convincingly showed that the two heads have distinct stepping kinetics, which is affected separately by the tail. The manuscript is definitely worth publication in LSA. This is the first dual color tracking study of kinesin, so it also presents a technical advance in the field.

I am listing my comments below to help authors prepare a revised manuscript:

1. What is ATP concentration? Mention this in the main text.

Response: We now included the requested information (p.5 l.20).

2. Page 7, line 12: double exponential decay is usually used for the addition of two double exponential decay functions with separate amplitudes and rates. The authors refer to a convolution of two exponentials, in which two exponential decay functions with the same amplitude are subtracted from each other. At the limit where the rates become equal, this equation becomes the Gamma function. I suggest the authors use "convolution of two exponentials" as this is the right term to evaluate the kinetics of two sequential rate-limiting steps that lead to an observable step.

Response: We have now up-dated the manuscript accordingly (p.7 l.2,4,16,20&26, p.8 l.5,14,18, p.10 l.1,4, p.21 l.2,4,18, p.25, p.26).

3. Page 7, line 14: I am surprised to see that the authors have not made the most out of their dcFIONA data. Do the heads strictly walk hand-over-hand? Does the stepping originate from the trailing position, and end up in a leading position? How did they register the two fluorescent channels?

Response: We have actually not performed a mapping of one channel to the other. We register the two channels on the same emCCD chip in order to avoid any irregularities in the temporal relation of the data. A mapping of the two channels would allow to answer these questions and will be on our agenda for upcoming experiments.

4. Page 7 line 14-16 is vague. Explain more clearly that single color imaging does not detect when the unlabeled head takes a step and its kinetics follows convolution of two rate-limiting steps. If both heads are being labeled, this is no longer the case, and one can detect the stepping time of both heads. Therefore, two color traces can be broken down to detect stepping time of each head, and dwell time between the steps would follow a single exponential decay. This was mentioned in the next paragraph, but it would be more appropriate to mention it all together in a single paragraph.

Response: We explained the situation in more detail and also adjusted the next paragraph in order to be more comprehensive (p.7 l.15-23&26).

5. Page 2 line 24: Convolution of two exponentials is not fully convincing, especially in B. The authors need to run F-test to justify that addition of an extra parameter to the fit is justified by the fit itself. The best way to address this would be to lower ATP further or increase the frame rate, in order to detect the initial rise of the convolution function.

Response: We performed further statistical analysis of the fits. There are many different implementations of the F-test, we opted to look at the individual p-values of the coefficients in our fit. These values report the significance of the coefficient for the result of the fit. We report the results in Supplementary Figure 5 showing, that k_2 is important for the fits to KLP11 at the 99% confidence level (p.9 l.1-3, p.25).

6. The left side of Figure 3 is very confusing. Selection of dwell times longer than 2 s is arbitrary and cannot be justified. The right way to do this would be to fit the convolution function and compare those rates from the fit.

Response: The text now explains more clearly why we select dwell times that are longer than 2 s. Unfortunately, the fits of the parameters of the convolved exponentials are mostly dependent on the ratio of the two parameters, not so much on the actual values. In order to obtain more stable fits for the parameters for the long dwell times we selected the dwell times that do not show the characteristic feature of the convolution (p.10 l.1-5).

7. Page 9, line 18: It is confusing to refer to distinct rates as the first rate and the second rate. What do the authors mean? Are these faster and slower rates from the fit?

Response: We refer to the two rate constants that are of the same scale in the KLP11 dwell time fits. We rephrased this to emphasize that the two rates do not have any 'order' (p.8 l.23, p.9 l.11, p.11 l.9&11).

8. Page 10, line 5: This conclusion is an overstatement. The authors say that they speculate on a model in a previous page. But here they claim that they provide compelling support. They need to soften this conclusion as the model still remains speculative.

We rephrased the manuscript to reflect our reviewer's point. However, we would like to emphasize that even though this work does not present a full mechanistic framework, especially now in light of our new data (Suppl. Figure 4), we expose for the first time the inhibitory influence of the stalk/tail on the stepping of a kinesin motor. We tried to make this point clearer. We are of course happy to rephrase if deemed necessary (p.12 l.1-3).

9. Figure 1 legend: The formula used refers to lambda, which is often used as the lifetime of exponential decay, whereas the authors refer to rates (k) in the main text. In addition, lambda is multiplied with x, instead, they should have used t, the symbol of time. So, lambda*x should be replaced with k*t to avoid confusion. The authors should also report the number of measurements (number of motors and steps analyzed) and report the errors from the fit for each dataset.

Response: We adjusted the names of the parameters as suggested and added the information for the number of included motors. The errors from the fit are large for some parameters: 1) big values for the 'quasi' single exponential as the fit does not depend much on the actual value of this 2). The values in the fit that converges to a convolution of two exponentials because the fit does not depend on the actual values but on the ratio of the two. We state this in the Appendix and would like to not state the actual errors in the main text for better readability (p.7 l.5,9-11,13, p.8 l.19, p.20).

September 20, 2019

RE: Life Science Alliance Manuscript #LSA-2019-00456-TR

Dr. Zeynep Ökten
Technische Universität Muenchen
Department of Biophysics
James-Franck-Str.1
Garching D-85748
Germany

Dear Dr. Ökten,

Thank you for submitting your revised manuscript entitled "Resolving kinesin stepping: one head at a time". We would be happy to publish your paper in Life Science Alliance pending final revisions necessary, mainly to meet our formatting guidelines:

- Please revise your manuscript one more time to address the remaining criticism of reviewer #1
- The methods section is not detailed enough to allow others to perform similar analyses. Please be more explicit and include/share the code/algorithms used; please incorporate the suppl information (Fit to the convolution of two exponentials) into the main manuscript as well
- Please enter a summary blurb and a running title in our submission system
- Please upload all figures as individual files and without legends (legends should remain in the main ms text)
- Please upload the manuscript text in word docx file format
- Please list 10 authors et al. in the reference list
- Please add a callout in the ms text to Fig S4B and S4C

A. FINAL FILES:

- An editable version of the final text (.DOC or .DOCX) is needed for copyediting (no PDFs).
- High-resolution figure, supplementary figure and video files uploaded as individual files: See our

detailed guidelines for preparing your production-ready images, <http://www.life-science-alliance.org/authors>

B. MANUSCRIPT ORGANIZATION AND FORMATTING:

Sincerely,

Andrea Leibfried, PhD
Executive Editor
Life Science Alliance
Meyerhofstr. 1
69117 Heidelberg, Germany
t +49 6221 8891 502

e.a.leibfried@life-science-alliance.org
www.life-science-alliance.org

Reviewer #1 (Comments to the Authors (Required)):

The authors have not changed their figure to make it clear what construct they are using (in response to my point 1 and 3). I think this deliberately misleading the the reader.

Otherwise the authors have addressed my comments and the paper is ready for publication.

Reviewer #2 (Comments to the Authors (Required)):

In this revised manuscript, the authors answered most of the comments by this and other reviewers. Now this work would be suitable for the publication in LSA.

Point-by-point response

1. Please provide the reference list according to the 10 references et al. format

We now used the EMBO Reports citation style available for Mendeley.

2. Reviewer #1: The authors have not changed their figure to make it clear what construct they are using (in response to my point 1 and 3). I think this deliberately misleading the the reader.

In Figure 2C, we now show the tail present at the heads to make clear that this is a wild-type motor.

September 27, 2019

RE: Life Science Alliance Manuscript #LSA-2019-00456-TRR

Dr. Zeynep Ökten
Technische Universität Muenchen
Department of Biophysics
James-Franck-Str.1
Garching D-85748
Germany

Dear Dr. Ökten,

Thank you for submitting your Research Article entitled "Resolving kinesin stepping: one head at a time". It is a pleasure to let you know that your manuscript is now accepted for publication in Life Science Alliance. Congratulations on this interesting work.

DISTRIBUTION OF MATERIALS:

Again, congratulations on a very nice paper. I hope you found the review process to be constructive and are pleased with how the manuscript was handled editorially. We look forward to future exciting submissions from your lab.

Sincerely,
